# Authentication of *Hippophae rhamnoides* ssp. *sinensis* and ssp. *mongolica* Based on Single Nucleotide Polymorphism at Ribosomal DNA and Their Vitamin Content Analysis

**DOI:** 10.3390/plants11141843

**Published:** 2022-07-14

**Authors:** Xiangmin Piao, Padmanaban Mohanan, Gokulanathan Anandhapadmanaban, Jong Chan Ahn, Jin Kyu Park, Deok Chun Yang, Gi-young Kwak, Yingping Wang

**Affiliations:** 1State Local Joint Engineering Research Center of Ginseng Breeding and Application, Jilin Agricultural University, Changchun 130118, China; pxm52_@163.com; 2Graduate School of Biotechnology, College of Life Sciences, Kyung Hee University, Yongin-si 17104, Korea; padmanabanm@khu.ac.kr (P.M.); cagokulanathang@hormail.com (G.A.); jongchanahn7@gmail.com (J.C.A.); pjinkyu53@gmail.com (J.K.P.); dcyang@khu.ac.kr (D.C.Y.)

**Keywords:** vitamin tree, SNP marker, subspecies authentication

## Abstract

*Hippophae rhamnoides* widely known as sea buckthorn berries (SB) are rich in vitamins and phytonutrients. The subspecies ssp. *sinensis* and ssp. *mongolica* are highly valued for their medicinal properties and vitamin contents, hence domesticated widely across Eurasia and Southeast Asia. Due to the frequent usage of these two subspecies, accurate identification is required to prevent economically motivated adulteration. In this study, we report the single nucleotide polymorphism (SNP) based molecular markers to easily distinguish these two subspecies at 45S nrDNA region. From the determined 45S rDNA region, we designed two primers (5′ sinensis and 5′ *mongolica*) and developed a multiplex PCR profile. The developed primers effectively distinguished the sea buckthorn subspecies in commercial products as well. Along with the development of subspecies specific primers, we have profiled vitamin contents from *H. rhamnoides* ssp. *sinensis* and ssp. *mongolica* and found ascorbic acid and riboflavin contents were high in both ssp. *sinensis* and spp. *mongolica*, yet the content of folic acid was high only in ssp. *mongolica*. Thus, we provide species specific primers and vitamin profile as an effective authentication of *H. rhamnoides*.

## 1. Introduction

*Hippophae* L. (Family: Elaeagnaceae), as sea buckthorn (SB), is a deciduous shrub producing berries enriched with vitamins. This genus is highly distributed from Northern Europe to Central Asia including, Himalayas and *H. rhamnoides* L. has widespread distribution throughout Asia and Europe [1,2]. The *Hippophae* genus contains seven species and eight subspecies. All of the seven described species and eight subspecies among eleven subspecies of *Hippophae* were determined to be distributed mainly from Heng-duan mountains to Qinghai-Tibet Plateau [3,4,5]. The identified species and subspecies have excellent stress resistance, water conservation, reforestation capabilities, but most significantly, their yellow-orange berries are recognized for their economic importance, hence domesticated in many countries [6].

The endemic growth of *H. rhamnoides* ssp. *sinensis* is on sandy soil by the riverbanks or along the river beds of mountainous slope with their habitat ranging from Qinghai to Hebei provinces and Sichuan to inner Mongolia. The latest systematic analysis of *Hippophae* species has reported that among the ssp. *sinensis*, ssp. *mongolica*, ssp. *turkestanica* and ssp. *yunnanensis*, are distributed in central and eastern Asia [3]. Sea buckthorn is a pioneer resource plant in China for water conservation, ecological restoration, most importantly for reforestation purposes [7]. The diverse habitats with various geographical altitude influences the nutritional composition and fruits shape of SB subspecies (Sun K 2006). The geographically distinct SB population was recently proved to have reasonable architectural, phenotypical and biochemical variations [8] that are due to genetic changes [9]. However, the fruits of native species have low economic value and poor agronomical traits. Due to which, the cultivation of sea buckthorn has undergone a dramatic change of introducing pharmaceutically and economically valued elite cultivars, such as ssp. *mongolica* (from Russia and Mongolia) and ssp. *sinensis* were introduced in northern China for cultivation and breeding purposes [10].

An enriched nutrition and bioactive compounds of the fruits and leaves of *Hippophae* species are used in food production, pharmaceutics and health care products [5,11]. A wide range of variations in Vitamin C, carotene, flavonoid concentrations exists among different genotypes and populations of *Hippophae* species [12,13]. Hence, the preservation of the subspecies with agronomic trait and phytonutrients within a population is important. Though the different ssp. of *H. rhamnoides* were classified, their phenotypical, biochemical and genetic differentiation remains elusive. Efforts were made to distinguish subspecies via the randomly amplified polymorphic DNA (RAPD) [4,14,15,16], inter simple sequence repeats (ISSR) [15,17], simple sequence repeats DNA (SSR) markers [18], and amplified fragment length polymorphism (AFLP) [19]. However, the mentioned markers lack the reproducibility and accuracy due to their random amplification by short primers, importantly they require expert handling and analysis of the results. However, the single nucleotide polymorphism (SNP) based markers are advantageous due to their abundancy in genome, variation among individuals within the species, biallelic nature, high-throughput and accessibility [20].

PCR-SNP based marker system has been used to distinguish cultivars and species [21,22,23,24,25]. Our goal in distinguishing *H. rhamnoides* especially the ssp. *sinensis* and *mongolica* was reached with the strategic SNP marker design. We found many SNP loci in nuclear 45S rDNA region of SB ssp. *sinensis*, ssp. *mongolica*. By using these SNP polymorphic loci, we differentiated the sub species along with their quantification of different vitamin contents as an additional differentiation strategy. We believe that the strategic SNP marker and vitamin content variations will provide a useful tool in authentication of SB ssp. *sinensis* and *mongolica*.

## 2. Results

### 2.1. Hippophae rhamnoides Species Identification

The SB samples obtained from various places were carefully analyzed for their differences before proceeding further. An analogous berry arrangement was observed in SB1, SB2 and SB5 with bright yellow color in SB5. Similarly, SB3, SB4 and SB6 had analogous berry arrangement, with SB4 had a bright orange color than SB3 and SB6 (Figure 1A). To analyze the predominant subspecies used in South Korea, we obtained SB tree samples and included in our experiment (Table 1, Figure 1B). The SB products had color variations, which initiated us to take them as our starting material (Figure 1C).

### 2.2. Sequence Analysis, Development of SNP Marker and the Authentication of SB Subspecies

To analyze the species of the samples in Figure 1, we used 45S rDNA ITS region targeting universal primers (Table 2). Upon, sequence analysis, we found a striking variation in the abundant SNP sites, thus we believed it could serve as a suitable region to distinguish SB. ssp. (Appendix A). The BLAST search at NCBI database has shown that, our samples were of SB. ssp. *sinensis* and SB. ssp. *mongolica* type, thus we have taken these two subspecies for further analysis. A 700 bp fragment of ribosomal 45S region was amplified with universal primers. We retrieved 45S rDNA sequences of SB. ssp. *sinensis* and ssp. *mongolica* from NCBI and performed multiple sequence alignments with our SB samples. We found an abundant SNP variation among our samples. There was a clear differentiation between ssp. *sinensis* and ssp. *mongolica*. Out of all the possible SNP sites, we found transversions such as, guanine (G) to thiamine (T) at 28th and 48th base pair and a transition from adenine (A) to guanine (G) at 42nd position is best suited to distinguish ssp. *sinensis* from ssp. *mongolica*. Thus, nucleotides between 24 to 48 bp in length were chosen to design primer 5′ sinensis (Figure 2A). Similarly, ssp. *mongolica* had SNP variations in regions flanking 175th bp to 198th base pairs and was chosen to design 5′ mongol primer (Figure 2B). There was a transition of cytosine (C) to thymine (T) at the 187th position and two transversions at the 191st and 198th positions as cytosine (C) to adenine (A) and guanine (G) to thymine (T), respectively. A common reverse primer 3′ SB was designed between the 641st to 660th base pairs to amplify these two forward primers (Figure 2C). We used all these primer pairs in a multiplex PCR reactions to authenticate SB. ssp. *sinensis* and *mongolica* (Figure 3A). The validation and authentication of SB samples were performed in a multiplex PCR reaction by using a combination of 5′ sinensis and 5′ *mongolica* primer pairs. All the samples such as SB berries (SBB), SB trees (SBT) and SB products (SBP) were used to validate the specific primers designed. The multiplex PCR profile produced an amplicon of 636 bp by the combination of primer pairs 5′ sinensis and 3′ SB for ssp. *sinensis* and a 489 bp-specific band for 5′ *mongolica* and 3′ SB primer pairs for ssp. *mongolica* (Figure 3B). The results were reproduced and tested several times as well as with various vitamin tree berry products as well. Thus, this primer pairs would distinguish the major subspecies of Hippophae easily.

### 2.3. Validation of SB Subspecies Specific Primers

The efficiency of ssp. *sinensis* and ssp. *mongolica* specific primers were tested against the concentration gradient of SB genomic DNA. We varied the DNA concentrations as 2 ng, 5 ng, 10 ng, 15 ng and 20 ng. The ssp. *sinensis* and ssp. *mongolica* specific primers could amplify the SB DNA at all concentrations (Figure 4A), thus validating their efficiency. Moreover, we obtained the dried SB berry samples from the local markets of South Korea and China (Table 1), which are difficult to distinguish based on appearance and morphology. We used our multiplex assay for the authentication of these samples. Our primer pairs identified these two subspecies easily. The samples SBB_D1, SBB_4, SBB_D5 produced bands for both that subspecies suggesting that they might had these two cultivars mixed, thus making a way for economically motivated adulteration (Figure 4B).

### 2.4. Riboflavin, Folic Acid and Ascorbic Acid Analysis form the SB Berries

SB berries are known for their abundant vitamin contents. AS shown in Figure 1A, the color variation of berries propelled us to analyze the vitamin profiles in them. The Riboflavin, folic acid and ascorbic acid content in SB berries of different origins were summarized in Figure 4. The five different types of berries ranged in color, yellow, orange or yellow-orange and in size from small to large. All the SB berry juices were analyzed within 48 h to prevent vitamin degradation. Processing effects and storage stability affects the vitamin availability in SB juice at 6 °C after 7 days and 25 to 40 °C after 7 days of storage [26]. It was reported that riboflavin was found in the pulp of SB at the concentration of 1.45 mg/100 g from Leh valley in Trans-Himalaya [27]. In our study, the riboflavin content was 0.16 mg gm^−1^ fresh weight (FW) in SBB #2 (ssp. *sinensis*) and 0.15 mg/g FW in SBB #4 (ssp. *mongolica*). The Folic acid content was the highest in SSB #3 (0.42 mg gm^−1^, ssp. *mongolica*), SBB #4 (0.35 mg gm^−1^, ssp. *sinensis*) and SBB #6 (0.30 mg gm^−1^, ssp. *sinensis*) compared to the others. We found both subspecies were enriched with ascorbic acid content, whilst SSB #2 (ssp. *sinensis*) contains 6.06 mg gm^−1^ FW (Table 3). Thus, based on our analysis both the subspecies have shared a high vitamin contents, however it was nearly impossible to distinguish them based on their vitamin profile. To our knowledge, this is the first report of simultaneous determination of these three vitamins by HPLC-VWD from SB berries fresh juice.

## 3. Discussion

*H. rhamnoides* (Sea buckthorn) is an economically important plant species that are domesticated worldwide. The flesh of SB berries contains a diverse complex of vitamins, mineral substrates, amino acids, organic acids, terpenoids and other phytochemicals [2]. *H. rhamnoides* subspecies are sometimes misidentified due to their similarities in vegetative morphology, their misidentified and mislabeled fruits are sold as dried forms or as powder [5]. Hence, it is necessary to properly identify different medicinally important *H. rhamnoides* subspecies. With the advent of high PCR amplification efficiencies and improved molecular discrimination methods, the development of simple, accurate DNA marker system had gained popularity. Thus, by using 45S rDNA region, we successfully developed a SNP based molecular marker for the authentication of *H. rhamnoides* subspecies. We found numerous SNP sites in 45S rDNA region for ssp. *sinensis* and ssp. *mongolica*. The transition and transversions of SNP mutations were abundant at this region (Appendix A). SNP based markers were used from a decade ago to rapidly identify the genomic variation. Their abundance in the genome has enabled as the marker system of choice [28]. Based on the multiple alignments of different SB samples, we designed 5′ sinensis and 5′ mongol and 3′ SB primers to differentiate ssp. *sinensis* and ssp. *mongolica* (Figure 2A–C). Recently, Liu Y et.al produced a high resolution melting based DNA barcoding method to authenticate seven different native Chinese *Hippophae* species using ITS2 region [29]. The species specific primers were validated for distinguishing the subspecies (data not shown) before multiplex PCR. Thus, we developed a multiplex PCR method to distinguish these subspecies in a single PCR by using the combination of these primers (Figure 3A). Our multiplex PCR amplification produced a 636 bp fragment for ssp. *sinensis* and 489 bp sized fragment for ssp. *mongolica* (Figure 3B). The SNP markers were unique and as predicted, they authenticated subspecies without any cross amplification. The SNP marker primers were effective for the SB products and identified the subspecies correctly. Thus, we could suggest that the single nucleotide polymorphic sites at 45S rDNA region serves as the dominant markers to distinguish these subspecies. The 45S rDNA genes in plants were transcribed by RNA polymerase I to 18S, 5.8S, and 28S rRNAs as the most convenient target for the molecular differentiation due to their small size generally between 600 to 800 bp. Their multi-copy nature makes amplification possible even with a low quantity of DNA or even with a highly degraded sample, and the ITS region usually has high similarity between the species [30,31,32]. Additionally, we tested the efficiency of our primers against minimum level of subspecies identification using a concentration gradient. Based on our results, we could suggest that our primers effectively identified these subspecies even with the DNA concentration as low as 2 ng (Figure 4A). Our multiplex assay from the dried SB berry samples suggests, that the PCR amplification and authentication could be possible. Moreover, we could identify the sub species using our multiplex assay (Figure 4B). Furthermore, we might assume that these sub species of *H. rhamnoides* could be popular among others (Table 1; Figure 3 and Figure 4B). Due to the unavailability and difficulty of obtaining other SB samples, we would like to reserve our analysis to the future. Similar to the research done by Li H et al. [10], an extended, comprehensive analysis of sea buckthorn species and sub species is necessary to distinguish and identify EMA of SB berries and products. Subsequently, we analyzed the vitamin contents of SB berries by HPLC as well. Vitamins are classified into oil and water-soluble compound, based on their solubility and absorption properties. Riboflavin, Folic acid and ascorbic acid are water soluble in nature and they were considered for analysis in this study. The HPLC determination is challenging due to the polarity that vitamins have especially in the fresh SB berry juice samples, but for these three vitamins, we found that the vitamins can be separated easily through our HPLC system specifications and mobile phase. Stability is highly recommended for better quantification in HPLC vitamin analysis, hence the sample processing and storage was taken care of with much precision. Comparatively, our vitamin reports are similar in other studies, but the total amount was varied due to the SB berries harvesting time. In our experiment, the contents of ascorbic acid and riboflavin were higher in ssp. *sinensis* (SB2) as well as in ssp. *mongolica* (SB4), respectively, while these plants were originated in China and Mongolia, respectively. Folic acid is higher in ssp. *mongolica* (SB3); however, the folic acid content is higher only in the plants grown in China. In terms of ascorbic acid, ssp. *sinensis* (SB2) had the highest quantity, though the plants introduced from Mongolia (SB4, SB5, SB6) have retained their ascorbic acid content (Figure 5; Table 2). As previously reported, the variations in vitamin contents in ssp. *sinensis* and ssp. *mongolica* berries could possibly be attributed to subspecies characteristics, their level of maturity and geographical locality including soil type [33,34]. Nevertheless, the relationship between adaptation, primary or secondary metabolism of these subspecies needs to be investigated further.

## 4. Materials and Methods

### 4.1. Sample Collection

Fifteen different SB berry samples were obtained from various places in Korea, China, Inner Mongolia and Mongolia for sequence analysis and HPLC analysis. They were procured from our collaborators in various locations in China including Inner Mongolia and Mongolia. A detailed description of samples used for the experiment is given in Table 1.

### 4.2. DNA Isolation and Sequence Analysis

The fresh samples were frozen in liquid nitrogen and ground to a fine powder with mortar and pestle, while the SB products were directly used in genomic DNA extraction and purification according to manufacturer’s instructions (Exgene Cell SV Kit, GeneAll, Daejon, Korea). By using the universal 45SF and 45SR (Table 2) primer pairs at the 45S ITS region [35] of SB was amplified. Polymerase chain reaction (PCR) was carried out in 20 μL volume using 20 ng of template DNA, 0.5 μM concentration of primers and 10 μL of 2X PCR premix (Genotech, Daejon, Korea). The amplification profile consisted of denaturation at 94 °C for 5 min, followed by 35 cycles of 94 °C 30 s, 60 °C 30 s, 72 °C for 1 min with a final extension of 72 °C for 10 min. The PCR products were separated by 1.5% agarose gel electrophoresis and documented. The PCR products were purified with PCRquick-spinTM (iNtRON Biotechnology, Daejeon, Korea) according to the manufacturer’s instructions and sequenced. The sequences were aligned using BLAST program in the NCBI database, highly similar sequences were compiled and edited using BioEdit program [36].

### 4.3. Design of Specific Primers for the Authentication of SB Subspecies

To differentiate SB subspecies, two specific primers were designed based on the SNP regions at 45S rDNA region in *H. rhamnoides* ssp. *sinensis* and ssp. *mongolica*. Parameters such as primer lengths from 21 to 25 nucleotides, and GC content from 30% to 50% were strictly followed for efficient amplification. Primer pairs 5′ sinensis and 5′ mongol were used to differentiate *H. rhamnoides* ssp. *sinensis* and ssp. *Mongolica*, respectively, and 3′ SB was used as a common reverse primer. Primer sequences, annealing temperature, priming regions and respective subspecies are listed in Table 3.

### 4.4. Sample Preparation for Vitamin Analysis

Healthy SB berries without any damages were used for the analysis. First, 100 g of healthy berries were homogenized with a Philips HR2860 mini blender and centrifuged at 1500 rpm for 15 min and the crude extract was filtered by commercial filters. The resulting berry juice concentrate was freeze dried and stored at 2 °C until the vitamin analysis.

### 4.5. Analysis of Riboflavin, Folic Acid and Ascorbic Acid of SB Berries by HPLC

The amount of the vitamins in SB berries were determined by high performance liquid chromatography (HPLC) analysis as previously described with some modifications [24]. Briefly, HPLC-VWD coupled with gradient pump was carried out using an Agilent 1260 Infinity II binary system (CA, USA). The filtered berry extracts were resolved on a column C18 (Kinetex ID 2.6 μm, 50 mm × 4.6 mm), with 25 mM Phosphate buffer and (solvent A) and acetonitrile (solvent B) at A: B ratios of 97:3, 20:80 and 97:3 with run times of 3, 18 min and 50 s, and 20 min, respectively, at a flow rate of 1.0 mL/min and detection wavelength of 280 nm.

## Figures and Tables

**Figure 1 plants-11-01843-f001:**
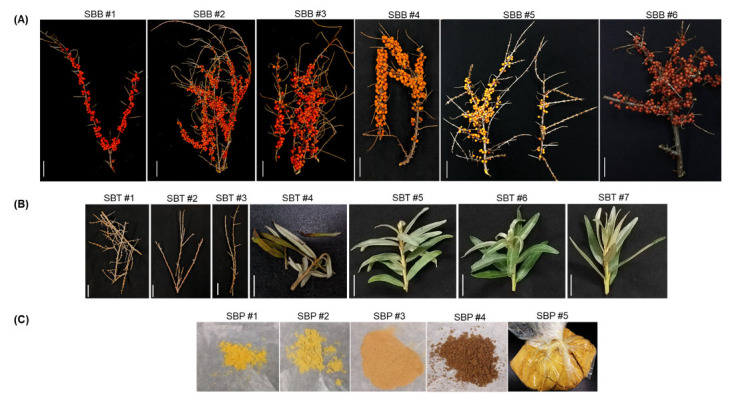
*H. rhamnoides* sub species samples used in this study. (**A**) SBB—Sea buckthorn berries; (**B**) SBT—Sea buckthorn tree; (**C**) SBP—Sea buckthorn products.

**Figure 2 plants-11-01843-f002:**
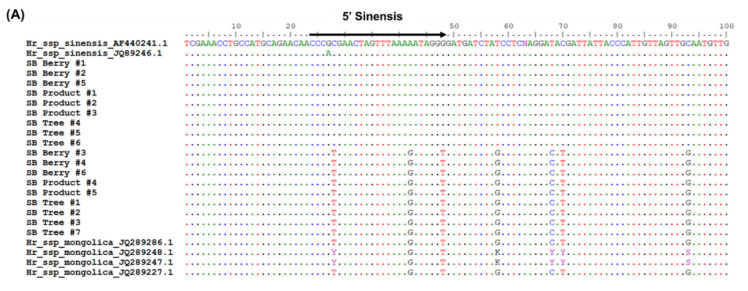
Multiple alignment of SB. ssp. *sinensis* and ssp. *mongolica* 45S rDNA region and position of (**A**) SB. ssp. *sinensis* specific (**B**) SB. ssp. *mongolica* specific SNP marker primers and (**C**) SB common reverse primer.

**Figure 3 plants-11-01843-f003:**
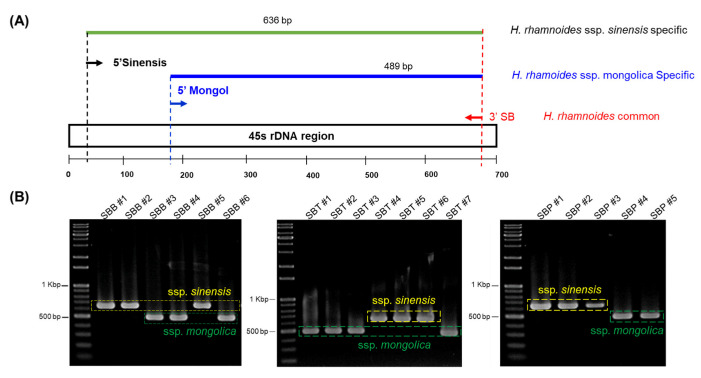
Authentication of SB subspecies *sinensis* and *mongolica*. (**A**) Schematic representation of SB 45S rDNA region and marker primer design with amplicon length. (**B**) Multiplex PCR gel pattern for SB. ssp. *sinensis* and ssp. *mongolica* by the combination of sub species specific primer pairs. SBB—Sea buckthorn berries; SBT—Sea buckthorn tree; SBP—Sea buckthorn products.

**Figure 4 plants-11-01843-f004:**
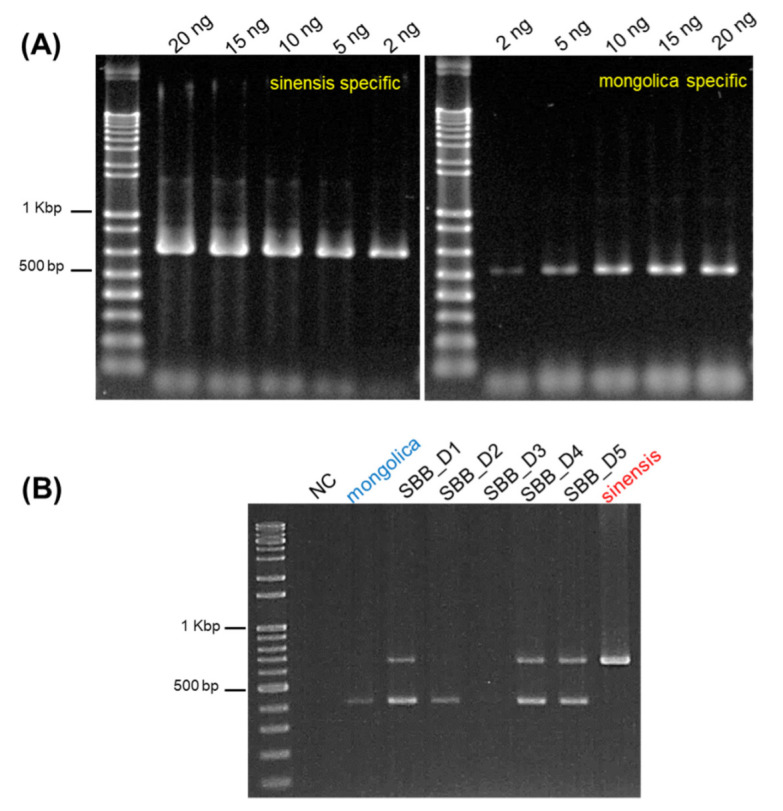
Validation of SB subspecies primers. (**A**) Concentration dependent amplification of ssp. *sinensis* and ssp. *mongolica* specific primers. (**B**) Multiplex PCR gel pattern for SB ssp. *sinensis* and ssp. *mongolica* by the combination of sub species specific primer pairs using dried SB berry products. NC—negative control; SBB_D—SB berries dried.

**Figure 5 plants-11-01843-f005:**
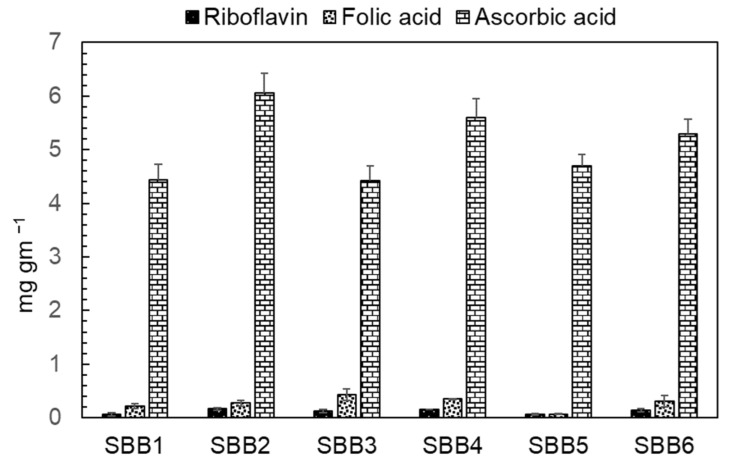
Vitamin content analysis from SB ssp. *sinensis* and ssp. *mongolica*.

**Table 1 plants-11-01843-t001:** Samples used in this study. SBB—Sea buckthorn berries, SBT—Sea buckthorn tree; SBP—Sea buckthorn products; SBB_D—Sea buckthorn berries dried.

Name	Plant Origin	Place of Purchase	Identified as
SBB1	China	China	*H. rhamnoides* ssp. *sinensis*
SBB2	China	China	*H. rhamnoides* ssp. *sinensis*
SBB3	China	China	*H. rhamnoides* ssp. *mongolica*
SBB4	China, but plants introduced from Mongolia	Local market Inner Mongolia	*H. rhamnoides* ssp. *mongolica*
SBB5			*H. rhamnoides* ssp. *sinensis*
SBB6			*H. rhamnoides* ssp. *mongolica*
SBP1	China	Local Market China	*H. rhamnoides* ssp. *sinensis*
SBP2	Mongolia	Local Market China	*H. rhamnoides* ssp. *sinensis*
SBP3	Obtained in South Korea (originated from Tibetan region)	e-commerce, South Korea	*H. rhamnoides* ssp. *sinensis*
	Mongolia		*H. rhamnoides* ssp. *mongolica*
SBP5	Inner Mongolia, China	Local marketInner Mongolia	*H. rhamnoides* ssp. *mongolica*
SBT1	South Korea	Obtained in South Korea	*H. rhamnoides* ssp. *mongolica*
SBT2			*H. rhamnoides* ssp. *mongolica*
SBT3			*H. rhamnoides* ssp. *mongolica*
SBT4			*H. rhamnoides* ssp. *sinensis*
SBT5			*H. rhamnoides* ssp. *sinensis*
SBT6			*H. rhamnoides* ssp. *sinensis*
SBT7			*H. rhamnoides* ssp. *mongolica*
SBT8			*H. rhamnoides* ssp. *mongolica*
SBB_D1	Not available	South Korea and China	Mixture of both ssp. *sinensis* and ssp. *mongolica* berries (?)
SBB_D2			*H. rhamnoides* ssp. *mongolica*
SBB_D3			Not amplified
SBB_D4			Mixture of both ssp. *sinensis* and ssp. *mongolica* berries (?)
SBB_D5			Mixture of both ssp. *sinensis* and ssp. *mongolica* berries (?)

**Table 2 plants-11-01843-t002:** Primers used in this study.

Primer	Primer Sequence	Tm (°C)	Amplicon Size (bp)	Target
45SF	GCGAGAATTCCACTGAACCT	60	800 bp	45S rDNA
45SR	ACGAATTCCCTCCGCTTATTGATATGCTTA	60		ITS region
5′ sinensis	CCCACGAACTAGTTTAAAAATAGGG	60	636 bp	SB. ssp. *sinensis*
5′ mongol	CGCAGATCGCGTCAAGGAACTAT	59	489 bp	SB. ssp. *mongolica*
3′ SB	ATGCCTCTTGATGCGACCCC	62		SB commonreverse

**Table 3 plants-11-01843-t003:** Vitamin contents in SB berries.

Sample Name	Riboflavin (mg gm^−1^)	Folic Acid (mg gm^−1^)	Ascorbic Acid (mg gm^−1^)	Sub-Species Type
SBB #1	0.07053	0.21809	4.430964851	ssp. *sinensis*
SBB #2	0.16518	0.27066	6.061464153	ssp. *sinensis*
SBB #3	0.12042	0.42719	4.421275605	ssp. *mongolica*
SBB #4	0.15491	0.35373	5.59301676	ssp. *mongolica*
SBB #5	0.05881	0.06914	4.693505587	ssp. *sinensis*
SBB #6	0.13491	0.30373	5.29301676	ssp. *mongolica*

## Data Availability

Not applicable.

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
