# Peer review of "Authentication of Hippophae rhamnoides ssp. sinensis and ssp. mongolica Based on Single Nucleotide Polymorphism at Ribosomal DNA and Their Vitamin Content Analysis"

_plants, 2022, doi:10.3390/plants11141843_

Round 1

Reviewer 1 Report

The authors designed the primers to 45S r DNA region  which allow to amplify and to sequence  target sequences with  several  SNP  distinguishing between two subspecies of Hippophae ramnoides. Also the content of 3 vitamins was studied and the difference between subspecies was found.

The paper has scientific and practical value and can be accepted after minor revision.

I have several minor comments.

Hippophae – in italics

Subspecies names  sinensis and mongolica – in italics

Sea buckthorn  -  sea with a capital or with a small letter – it is desirable to use I variant in the text.

Line 15 Hippophae rhamnoides  as sea buckthorn barriers (SB) are rich in vitamins

It is not clear what does "barriers" mean here?

Line 54 A wide range of variations in Vitamin C, carotene – it is not clear why sometimes “vitamin” with a capital  letter (also in Lines 130, 131).

Line 90   We used universal primers in the Chloroplast and 45S rDNA regions.

Chloroplast -??

It is necessary to cite the work in which the universal primers 45S rDNA were designed (Jobst et al., 1998),  and  to report to which parts of the 45s rDNA the primers are complementary.

JOBST, J., RING, K., HEMLEBEN, V., 1998: Molecular evolution of the internal transcribed spacer (ITS1 and ITS2) and phylogenetic relationships among species of the family Cucurbitaceae. - Molec. Phylogenet. Evol. 9: 204-219.

Line 133  The five different berries having distinct color from yellow, orange and yellow-orange which varied from size between small and large.

The five different types of berries ranged in color, yellow, orange or yellow-orange, and in size from small to large.

Line 164  SNP based markers were developed recently, mainly due to their

abundance in the genome and for better resolution [27].

Still, SNPs began to be used for taxa identification much earlier than 2017.

cf.  Ganal et al. SNP identification in crop.  Curr Opin Plant Biol . 2009 12(2):211-7.

Line 198  As previously reported, the variations in vitamin contents could possibly be attributed to subspecies characteristics, their level of maturity and geographical locality including soil type [32 - 33] in ssp. sinensis, ssp. mongolica berries.

As previously reported, the variations in vitamin contents in ssp. sinensis, ssp. mongolica berries could possibly be attributed to subspecies characteristics, their level of maturity and geographical locality including soil type [32 - 33].

Please check attentively using of italics in taxa names and in the journal titles  in the References (N 6, 17).

Number 7 in the References is empty

I also made some corrections in the pdf file of the MS.

Author Response

The authors designed the primers to 45S r DNA region  which allow to amplify and to sequence  target sequences with  several  SNP  distinguishing between two subspecies of Hippophae ramnoides. Also the content of 3 vitamins was studied and the difference between subspecies was found. The paper has scientific and practical value and can be accepted after minor revision.

We thank the reviewer for the appreciation towards our MS. We also value the time taken by the reviewer for his critical suggestions, and corrections. We made the corrections as per the reviewer comments in the revised MS. The corrections were highlighted for easier understanding

I have several minor comments.

Hippophae – in italics

Subspecies names  sinensis and mongolica – in italics

Sea buckthorn  -  sea with a capital or with a small letter – it is desirable to use I variant in the text.

Line 15 Hippophae rhamnoides  as sea buckthorn barriers (SB) are rich in vitamins

It is not clear what does "barriers" mean here?

There was a typo in this statement. The word "berries" was corrected and highlighted in the MS

Line 54 A wide range of variations in Vitamin C, carotene – it is not clear why sometimes “vitamin” with a capital  letter (also in Lines 130, 131).

Thank you for the correction. All the mistakes in the word were corrected and highlighted in the revised MS (Lines 133, 134)

Line 90   We used universal primers in the Chloroplast and 45S rDNA regions.

Chloroplast -??

We thank reviewer for finding out this mistake and the redundant and oversight statement was removed form the revised MS 

It is necessary to cite the work in which the universal primers 45S rDNA were designed (Jobst et al., 1998),  and  to report to which parts of the 45s rDNA the primers are complementary.

JOBST, J., RING, K., HEMLEBEN, V., 1998: Molecular evolution of the internal transcribed spacer (ITS1 and ITS2) and phylogenetic relationships among species of the family Cucurbitaceae. - Molec. Phylogenet. Evol. 9: 204-219.

We thank the reviewer for this valuable suggestions. The 45S ITS region universal primers were used in our research. In the revised MS it was mentioned and highlighted as well (lines. 93, 131, 218). The reference suggested was also included (line. 332).

Line 133  The five different berries having distinct color from yellow, orange and yellow-orange which varied from size between small and large.

The five different types of berries ranged in color, yellow, orange or yellow-orange, and in size from small to large.

We really appreciate the reviewer in taking the time out to share your corrections in the statements. We corrected the statement as you have suggested. The corrections are in Line 136-137 and highlighted as well.

Line 164  SNP based markers were developed recently, mainly due to their abundance in the genome and for better resolution [27].

Still, SNPs began to be used for taxa identification much earlier than 2017.

cf.  Ganal et al. SNP identification in crop.  Curr Opin Plant Biol . 2009 12(2):211-7.

We thank and appreciate the reviewer for the suggestions. We come across with this paper just now and we agree to replace the existing review with this new one. Duly the statement was also corrected according to the new reference (Line no. 167-169 and 317)

Line 198  As previously reported, the variations in vitamin contents could possibly be attributed to subspecies characteristics, their level of maturity and geographical locality including soil type [32 - 33] in ssp. sinensis, ssp. mongolica berries.

As previously reported, the variations in vitamin contents in ssp. sinensis, ssp. mongolica berries could possibly be attributed to subspecies characteristics, their level of maturity and geographical locality including soil type [32 - 33].

We thank the reviewer for this correction in our statement. As per the suggestions the corrections were included in the revised MS and highlighted (Line no. 202-205)

Please check attentively using of italics in taxa names and in the journal titles in the References (N 6, 17).

Number 7 in the References is empty

I also made some corrections in the pdf file of the MS.

We corrected all the mistakes and included the suggestions as per reviewer's comments

Reviewer 2 Report

The manuscript was grammatically correct and easy to read.  The methods were a little dated, but the utility of the work was directed to a quick assay to detect adulteration of the product.

However, the authors failed to show that their assay would be specific to only the subspecies sinensus and mongolica.  What is missing is any evidence that this assay would not incorrectly identify the other Hippophae species as spp. sinensus or mongolica.  What is needed is data showing that the sequences differences seen in sinensus and mongolica are specific to only those subspecies.  Amplification and sequence analysis of the other members of the Hippophae genus would prove that they are.  Since the stated purpose of this work was to come up with an easy to perform screen for adulterants of the product, it is incumbent on the authors to show that it is indeed specific to those subspecies.  

This may be an non-issue, biologically due to differences in the berries of the other species or lack there of, but the authors did not provide any descriptions of the other species to make a determination.  In short the background information was lacking due to this.

Author Response

We thank and appreciate reviewer for the concern towards our Manuscript. We also value the time taken by the reviewer for his critical suggestions, and corrections All the authors have accepted the revised MS to be submitted and have done the changes as per the reviewer comments in the revised MS. The authors replies to the reviewer’s comments as follows.

Reviewer: The manuscript was grammatically correct and easy to read. 

Authors Comments: We thank the reviewer and take this is as an appreciation for the comments shared.

Reviewer: The methods were a little dated, but the utility of the work was directed to a quick assay to detect adulteration of the product.

Authors Comments: We agree with the reviewer’s view on this ad thanks for the comment. Yes, the usage of this detection method is to enable the distinction of these medicinally valuable species / subspecies. Consequently, we opt for this straightforward method for the detection.

Reviewer: However, the authors failed to show that their assay would be specific to only the subspecies sinensus and mongolica. What is missing is any evidence that this assay would not incorrectly identify the other Hippophae species as spp. sinensus or mongolica. 

Authors Comments: It’s a critical point of our MS and thank the reviewer to explain our stance on this point. When we began this project, we wanted to find markers for all Hippophae species as well as subspecies. However, due to the predominance usage of H. rhamnoides ssp. sinensis and mongolica and due to their success in agronomical traits we couldn’t obtained materials from other species or subspecies of Hippophae in our study.

Recently Li H et al. (https://doi.org/10.1371/journal.pone.0230356) has done a diversity analysis of Hippophae rhamnoides accessions. According to their analysis, out of 78 accessions from northern China, 52 form ssp, mongolica and 6 from ssp. sinensis while the remaining samples were of hybrids between these two cultivars. We attach here, the list of materials used in their research for your perusal as well (Table 1).

Thus, due to the lack of materials of other subspecies, we were unable to show our markers specificity with other subspecies.

Table 1 summarizes the information of the plant materials used  (From Li H et al., 2020, PLOS ONE 15(3): e0230356.)

Reviewer: What is needed is data showing that the sequences differences seen in sinensis and mongolica are specific to only those subspecies.  Amplification and sequence analysis of the other members of the Hippophae genus would prove that they are.  Since the stated purpose of this work was to come up with an easy to perform screen for adulterants of the product, it is incumbent on the authors to show that it is indeed specific to those subspecies. This may be an non-issue, biologically due to differences in the berries of the other species or lack thereof, but the authors did not provide any descriptions of the other species to make a determination. In short the background information was lacking due to this.

Authors Comments: We would like to address this concern concomitantly with the previous comment. As we mentioned, the original aim of this project the lack of information as well as the samples made us to use the predominant subspecies from Hippophae. With most of the cultivars, breeding population and products, produced form these subspecies we believe our work will significantly prevent economically motivated adulteration. The easy to perform SNP based identification would save considerable time for the breeders and protects consumer rites of taking precise pharmaceutics.

We added the following lines in our revised MS to support our findings.

Line number. 51-55: However, the fruits of native species have low economic value and poor agronomical traits. Due to which the cultivation of sea buckthorn has undergone a dramatic changes of introducing pharmaceutically and economically valued elite cultivars, such as ssp. Mongolica (from Russia and Mongolia) and ssp. sinensis were introduced in northern china for cultivation and breeding purposes [10].

Round 2

Reviewer 2 Report

If the sentence you added is meant to indicate that you could visually detect a morphological difference in the berries of non-H. ramnoides species then that is helpful.  But what if all you have is a powder as shown in your Fig. 1, the powder could have come from any SB species that has colored berries. Since your assay uses PCR, you probably could amplify a product from a DNA extraction of the powder.  Without, some support from an assay of other members of the genus, could you be sure that you do not have an adulterated product?  Without evidence for specificity, your assay is of questionable utility. According to the text, "This genus is highly distributed from Northern Europe to Central Asia including, Himalayas".  This means that other members of the genus must be obtainable outside of China.  Even if the collection was not a complete representation of the genus, it would be better than nothing.

If the point of the paper is that you have an assay that distinguishes two subspecies of H. ramnoides, then you have presented enough evidence. But, if the point is that you can identify any other adulterating species of Hippophae then you have not presented enough evidence.

Because of the lack of either amplification and/or sequence from other Hippophae spp., the sentences in lines 23 and 26 are not correct, you have not presented evidence that your primers are specific to H. ramnoides. The evidence presented only says that you can amplify a  unadulterated product from H. ramnoides ssp. sinensis or mongolica and identify which subspecies the product came from. Additionally, it does not show that the PCR product's length or sequence from other species could not be confused with H. ramnoides ssp. sinensis or mongolica.

Author Response

Dear reviewer

We the authors of the

MS Title: Authentication of Hippophae rhamnoides ssp. sinensis and ssp. mongolica based on single nucleotide polymorphism at ribosomal DNA and their vitamin content analysis

Journal: Plants

appreciate your patience during our extended revision period. Your concern were addressed in the revised MS. The detailed response form all the authors were as follows

Reviewer: If the sentence you added is meant to indicate that you could visually detect a morphological difference in the berries of non H. rhamnoides species, then that is helpful. But what if all you have is a powder as shown in your Fig. 1, the powder could have come from any SB species that has colored berries. Since your assay uses PCR, you probably could amplify a product from a DNA extraction of the powder. 

Response: To our knowledge and so far in this project, most of the SB products or berries used commercially are majorly derived from H. rhamnoides ssp. sinensis and ssp. mongolica. We confidently could say this statements based on the results from Fig. 3, Fig. 4 and Table. 1. To justify this statement, research from Li H et. al 2020, PLOS one (https://doi.org/10.1371/journal.pone.0230356) has shown the diversity of SB within these two subspecies while they are genetically distinct.

The materials we collected from South Korea and China all belongs to these two cultivars and or the mixtures of them (Fig. 4B). During the extended second revision period we obtained more SB products, to extend our analysis. As the reviewer 2 pointed out, the dried berries we obtained did contain the mixture of ssp. sinensis and ssp. mongolica. The designed markers identified these two subspecies in the dried berries by amplifying the specific bands as shown in Fig 4B. Thus, our markers could authenticate these two subspecies easily even in the samples when the appearance, morphology of the berries are compromised. The detailed results, figure 4 and discussions were inserted in the revised MS in lines 135-144, 199-210.

Reviewer:  Without, some support from an assay of other members of the genus, could you be sure that you do not have an adulterated product?  Without evidence for specificity, your assay is of questionable utility. According to the text, "This genus is highly distributed from Northern Europe to Central Asia including, Himalayas". This means that other members of the genus must be obtainable outside of China. Even if the collection was not a complete representation of the genus, it would be better than nothing.

Response:  We agree with the reviewer’s point of view of including various members of the genus. However, we would be interested to employ the suggestion in our next project. As a fairly domesticated population of Hippophae genus, Hippophae rhamnoides species and the ssp. sinensis and ssp. mongolica are successfully domesticated due to the advantages they hold (as stated in the lines 51 – 55). Though the Hippophae genus is widely distributed, the domestications and cultivation of these members are hard. Moreover, we were unable to obtain all the members of SB to include in our analysis. Besides, we have shown that our markers could differentiate predominantly available subspecies sinensis and mongolica in breeding program and for commercial purposes. We would like to include reviewer’s suggestions in our future research.

Reviewer:  If the point of the paper is that you have an assay that distinguishes two subspecies of H. ramnoides, then you have presented enough evidence. But, if the point is that you can identify any other adulterating species of Hippophae then you have not presented enough evidence. Because of the lack of either amplification and/or sequence from other Hippophae spp., the sentences in lines 23 and 26 are not correct, you have not presented evidence that your primers are specific to H. ramnoides. 

Response:  Yes, the purpose of this research to distinguish predominantly used subspecies of H. rhamnoides. The statements were corrected and highlighted in the revised MS

Response:  The evidence presented only says that you can amplify an unadulterated product from H. ramnoides ssp. sinensis or mongolica and identify which subspecies the product came from. Additionally, it does not show that the PCR product's length or sequence from other species could not be confused with H. ramnoides ssp. sinensis or mongolica.

Response: The berry color of SB is predominant in all the SB products, powders we bought from the local markets. Besides, our multiplex profiles have included the samples from SB stem, SB products (Fig 3B) and dried berries (Fig 4B), thus proving we could effectively distinguish these sub species. In addition, we performed DNA concentration gradient dependent amplification of ssp. sinensis and spp. Mongolica primers. As shown in Figure4 A, our primers could amplify even 2ng of DNA present in the SB products. More over the use of additional species in PCR profile were didn’t required in this research.

Round 3

Reviewer 2 Report

For your series of papers to make sense, you should have started with validating your assay (demonstrating that the primers were specific) then using them to determine if the raw product was adulterated. Thus, this manuscript should be the second, not the first.